# Bridge Therapy before Liver Transplant for Advanced Hepatocellular Carcinoma

**DOI:** 10.3390/medicina60061010

**Published:** 2024-06-20

**Authors:** Valentina Bianchi, Erida Nure, Carmen Nesci, Marco Maria Pascale, Gabriele Sganga, Salvatore Agnes, Giuseppe Brisinda

**Affiliations:** 1Emergency Surgery and Trauma Center, Department of Abdominal and Endocrine Metabolic Medical and Surgical Sciences, Fondazione Policlinico Universitario A Gemelli, IRCCS, 00168 Rome, Italy; valentina.bianchi@guest.policlinicogemelli.it (V.B.); carmennesci96@gmail.com (C.N.); gabriele.sganga@policlinicogemelli.it (G.S.); 2General and Transplant Surgery, Department of Abdominal and Endocrine Metabolic Medical and Surgical Sciences, Fondazione Policlinico Universitario A Gemelli, IRCCS, 00168 Rome, Italy; erida.nure@policlinicogemelli.it (E.N.); marcomaria.pascale@policlinicogemelli.it (M.M.P.); salvatore.agnes@policlinicogemelli.it (S.A.); 3Catholic School of Medicine “Agostino Gemelli”, 00168 Rome, Italy

**Keywords:** hepatocellular carcinoma, transarterial chemoembolization, radiotherapy, sorafenib, orthotopic liver transplant

## Abstract

Hepatocellular carcinoma is the most common primary liver tumor. Orthotopic liver transplant is one of the best treatment options, but its waiting list has to be considered. Bridge therapies have been introduced in order to limit this issue. The aim of this study is to evaluate if bridge therapies in advanced hepatocellular carcinoma can improve overall survival and reduce de-listing. We selected 185 articles. The search was limited to English articles involving only adult patients. These were deduplicated and articles with incomplete text or irrelevant conclusions were excluded. Sorafenib is the standard of care for advanced hepatocellular carcinoma and increases overall survival without any significant drug toxicity. However, its survival benefit is limited. The combination of transarterial chemoembolization + sorafenib, instead, delays tumor progression, although its survival benefit is still uncertain. A few studies have shown that patients undergoing transarterial chemoembolization + radiation therapy have similar or even better outcomes than those undergoing transarterial chemoembolization or sorafenib alone for rates of histopathologic complete response (89% had no residual in the explant). Also, the combined therapy of transarterial chemoembolization + radiotherapy + sorafenib was compared to the association of transarterial chemoembolization + radiotherapy and was associated with a better survival rate (24 vs. 17 months). Moreover, immunotherapy revealed new encouraging perspectives. Combination therapies showed the most encouraging results and could become the gold standard as a bridge to transplant for patients with advanced hepatocellular carcinoma.

## 1. Introduction

Hepatocellular carcinoma (HCC) is the most common tumor of the liver, the sixth most common in the world, and the third leading cause of cancer mortality [1]. Cirrhosis, chronic viral infections (HCV and HBV), alcohol abuse, non-alcoholic fatty liver disease, type II diabetes mellitus, obesity, and smoking are the most frequent risk factors for HCC [2,3]. Orthotopic liver transplant (OLT) is the most successful treatment for patients with HCC because it is a solution both for HCC and cirrhotic liver, which is a risk factor for the onset of new tumors [4]. In 1996, Mazzaferro et al. introduced the first criteria for OLT in patients with HCC. The so called “Milan Criteria” included a solitary lesion of <5 cm and up to three nodules with an overall diameter of <3 cm without microscopic vascular invasion or extrahepatic disease [5]. There was a notable improvement in post-transplant survival (>70% 5-year survival) and a decrease in the recurrence of HCC in patients undergoing OLT (below 15%) after the introduction of the Milan criteria. This is why these criteria were used as the worldwide standard selection criteria for OLT [6]. El-Serag et al. showed that only 12% of HCC patients were treated with a curative transplant or resection, with a 3-year survival of >50%. All the others were selected for palliative treatment or no treatment, with a 3-year survival lower than 10%. It could also happen that patients considered as suitable for an OLT were de-listed at a certain point because their tumor could grow and exceed the criteria [7]. Llovet et al. highlighted that a patient well selected for resective surgery (low bilirubin levels and absence of portal hypertension) presented a better 5-year survival than those undergoing an OLT (84% vs. 54%). This was due to the shortage of organs and dropping out from the list [8]. Three different strategies were introduced in order to avoid this problem:(1)the use of living donors or non-standard deceased donors.(2)the use of bridge therapy, such as liver resection or locoregional therapies (transarterial chemoembolization, TACE; transarterial radio embolization, TARE; radiofrequency ablation, RFA; percutaneous ethanol injection, PEI; percutaneous laser ablation, PLA; and microwave ablation, MWA).(3)widening of the eligibility criteria.

Yao et al. were the first to propose the introduction of new expanded criteria [9]. The University of California, San Francisco (UCSF) criteria are based solely on morphological parameters. These include a single lesion of ≤6.5 cm or a maximum of three nodules with the largest ≤ 4.5 cm and a total tumor diameter of ≤8 cm, in the absence of metastasis and macrovascular invasion. Patients included according to these criteria (60 patients, 86%) had an overall survival at 1 and 5 years of 90% and 75.2%, respectively. On the other hand, patients outside the UCSF criteria (10 patients, 14%) showed a survival at 1 year of 50% (*p* < 0.0005). The new criteria did not determine a significant decrease in survival [9]. Therefore, the Milan group [10] attempted to enlarge their criteria, creating a new strategy called the “up-to-seven criteria” (new Milan criteria): the number of tumors plus the sum of the tumor diameter (in cm) has to be ≤7, with no vascular invasion. The up-to-seven group presented a 5-year survival rate of 71.2%. Later, other studies confirmed that the up-to-seven criteria were a useful tool to evaluate the potential candidates with HCC for OLT [11]. According to Barcelona Clinic Liver Cancer (BCLC) staging [12], there are different disease settings: very early; early; intermediate; advanced; and terminal. They are based on evaluation of the performance status (PS), Child–Pugh score, nodule characteristics, portal hypertension, associated pathologies, pathological lymph nodes, distant metastases, and portal thrombosis. The very early stage is categorized by a single tumor of ≤2 cm or in situ, Child–Pugh A, and a PS of 0. The early stage consists of a single or up to three HCC lesions of <3 cm, a PS between 0 and 2, and Child–Pugh A-B. The intermediate stage is characterized by asymptomatic, large, or multifocal nodules without vascular invasion or distant metastasis, a PS between 0 and 2, and Child–Pugh A-B. The advanced disease presents portal vein involvement, extrahepatic and lymph nodes metastasis, and a PS between 1 and 2. The terminal stage is determined by a PS of >2 and Child Pugh C [12,13].

The aim of this study is to develop a review on the effectiveness of bridge therapy before OLT for HCC.

## 2. Materials and Methods

We included all studies evaluating the effectiveness of bridge therapy in the case of advanced HCC. Articles were searched in PubMed, Web of Science, and Cochrane. The search keywords were as follows: “transarterial chemoembolization” or “chemoembolization” or “TACE” AND “advanced hepatocellular carcinoma” or “advanced HCC” or “advanced liver cancer” or “advanced liver tumor” AND “sorafenib”. We selected 185 articles. After deduplication, we considered 92 articles. The references were screened. The selected articles had to match the following inclusion criteria. First of all, they had to be not only linked to the topic, but also show significant results in terms of survival and de-listing. Articles had to be written in English. Moreover, we only considered studies involving adult patients (age ≥ 18). We included 24 articles and excluded 68 (Figure 1).

The aim of this study was to evaluate if bridge therapy in advanced HCC improves overall survival (OS) and/or reduces dropout from the waiting list and how its effectiveness can be improved. Even if all the studies show relevant results, they could hide a certain bias. In fact, there is an inevitable heterogeneity because the cohorts described in the various articles are different in terms of grade, stage of disease, and proposed treatments. There was no bias, instead, neither for the diagnosis nor for the outcomes. In fact, diagnosis was achieved through imaging (CT or MRI) in all the studies. Similarly, the outcomes were checked with objective restaging methods (CT or MRI; histology of the explant) after a strict definition.

## 3. Bridging Therapies in Very Early Stage, Early Stage, and Intermediate Stage, According to BLCL

For patients within the conventional transplant criteria, bridging therapies are used to delay tumor progression and minimize the risk of de-listing [14]. There are different bridging and downstaging strategies: surgery (liver resection), TACE, TARE, RFA, PEI, PLA, cryoablation, and MWA. Each strategy has indications, advantages, and disadvantages [15]. A scheme of the treatments proposed for the different stages of HCC according to Child–Pugh grade is shown in Figure 2.

### 3.1. Surgery

Surgical intervention has the advantage of providing better control over the growth of the tumor. Other methods, instead, do not always achieve complete tissue necrosis. The removal of the tumor mass also allows for histology and analysis of genetic expressions and microsatellites, which often worsen prognosis and can require liver transplantation in a shorter time [16,17,18].

Surgery is indicated for patients with a very early stage according to the BLCL System [12], with a normal portal pressure and bilirubin [13]. On the other hand, surgery is contraindicated in the case of severe portal hypertension or liver failure [3,15].

### 3.2. Transarterial Chemoembolization (TACE)

TACE is an arterial infusion of a lipiodol solution, consisting of a chemotherapeutic agent carrier, an augmenter of antitumor effects through efflux into the portal vein, and a micro-vessel embolic agent (Figure 3). The procedure is more effective in nodules with good vascularization by large tributary arteries [3,19]. However, it is contraindicated in the case of renal disorder characterized by a low glomerular filtration rate (<30 mL/min) and in the case of a compromised portal flow [20]. In standard TACE, several chemotherapic drugs are added with lipiodol. Non-resorbable microspheres can be an alternative to conventional lipiodol [21,22].

So-called “drug-eluting beads” (DEB) were developed to maintain doxorubicin from solution in order to release it in a sustained manner. The amount of chemotherapeutic agents in the systemic circulation can be substantially reduced compared with lipiodol-based TACE. This sharply increases the local drug concentration. For such a reason, it is recommended in the case of intermediate stage with recurrences [21,23,24].

Kim et al. evaluated the efficiency of TACE vs. RFA in very-early-stage patients. They found no significative differences between these two groups, even though RFA had a better tumor response and delayed tumor progression [25].

Jin et al. estimated the results of HR and TACE for single large early-stage HCC. They stated that HR presented a better 5-year survival rate in the group treated surgically (65% vs. 17%, *p* < 0.01) [26]. Lee et al. conducted a large study that compared long-term survival after HR and TACE as the initial treatment for large solitary HCC (>5 cm). They concluded that TACE can be an alternative preliminary treatment for large single HCCs if HR is not reasonable, especially for cases with portal hypertension [27].

Nowadays, TACE is indicated in the intermediate stage if patients are asymptomatic and for large or multifocal HCCs without vascular invasion or extrahepatic localization [28].

### 3.3. Transarterial Radioembolization (TARE)

Radioembolization is a technique that involves the cannulation of the hepatic artery and its subsequent embolization with 90Y-loaded microspheres and 131I-labelled iodinated poppy seed oil. Treatment with 90Y has a low toxicity [29]. In fact, the concentration is higher in the tumor than in the normal liver tissue because of the predominant arterial supply of hepatic tumors by the arteries and not by the portal system. It is indicated in the intermediate stage for patients with multiple nodules, like TACE [15]. To date, TARE has not shown a clear survival benefit, except in patients with a diagnosis of portal vein thrombosis, compared with TACE and any other treatment [30].

### 3.4. Ablative Therapies

Ablation consists of direct contact between the tumor and a chemical (chemical ablation) or thermal energy to achieve necrosis. There are two types of thermoablative therapy: hyperthermic (i.e., RFA, MWA, and PLA) and hypothermic (cryotherapy). Chemical ablation is carried out with PEI or percutaneous acetic acid injection (PAI). It is used in early-stage cases for patients with associated diseases [29,31,32].

#### 3.4.1. Radiofrequency Ablation (RFA)

Ablation is carried out with a thin needle inserted through the abdominal wall until reaching the tumor, guided by imaging. The probe applies alternating electrical energy (radiofrequency energy) in the tumor tissue, generating heat in the nodule. This causes necrosis and the destruction of the cancerous cells. The dead cells eventually turn into a harmless scar [33,34]. RFA is used when nodules are <3 cm and there is no perivascular invasion [29]. It is potentially dangerous for patients with coagulation disorders, superficial lesions, or tumors adjacent to the gallbladder, main bile ducts, or intestinal loops [34].

#### 3.4.2. Percutaneous Ethanol Injection (PEI)

PEI determines dehydration, necrosis of the tumor tissue, and small-vessel thrombosis, determining tumor ischemia and destruction. It has a better effect for nodules ≤ 3 cm. Moreover, it is also suitable for cases with impaired clotting parameters or nodules in sites which make thermal ablation unsafe [35,36]. Branco et al. demonstrated how PEI is effective and safe before liver transplantation. In fact, completely necrotic tumoral tissue was observed in 64.3% of the cohort. The additional nodules and the volume of the main HCC nodule in the explanted liver were significantly lower in the group treated with PEI than in the control group (*p* = 0.002). Drop out due to HCC progression was registered in 4.8% and 8.5% of participants and recurrence in 5% and 6.2%, respectively [37].

#### 3.4.3. Cryoablation

Ultra-low-temperature (−20 °C) ablation exploits argon to achieve a tissue injury thanks to temperatures under the freezing point. It is used in early-stage tumors with a diameter of <3 cm and perivascular involvement. It can be performed with conscious sedation, so it is a good choice for patients with comorbidities who are not good candidates for anesthesia [38,39].

#### 3.4.4. Percutaneous Laser Ablation (PLA)

This procedure uses temperatures up to 150 °C which are reached through light absorption, leading to coagulative necrosis. It is more effective in HCC ≤ 3 cm, and it is available for patients with impaired coagulation. Data have demonstrated that PLA could be dangerous in sub-glissonian nodules or in tumors adjacent to the major bile ducts, the gallbladder, or the bowel loops [40].

#### 3.4.5. Microwave Ablation (MWA)

MWA is a thermal ablation solution for the trans-abdominal treatment of HCC. It seems to be more effective than RFA in tumors of ≥3 cm or those adjacent to the large vessels [41,42].

## 4. Bridging Therapies in Advanced Stage According to BCLC

The BCLC stage C includes symptomatic patients and/or cases with extrahepatic spread or vascular invasion.

Sorafenib is considered to be the gold standard treatment for these cases [19], but some other studies have suggested that radiotherapy, TACE, immunotherapy, and combinations of these therapies could also be used to improve the outcomes of these patients.

### 4.1. Sorafenib

There was no systemic chemotherapy to treat advanced HCC until 2009. The Sorafenib Hepatocellular Carcinoma Assessment Randomized Protocol (also called “SHARP”) was created by Rimassa et al. [43]. They stated that sorafenib is reliable for a better survival and time-to-progression in cases with advanced HCC. Sorafenib is a multikinase inhibitor of the platelet-derived growth factor receptor, vascular endothelial growth factor receptor, and RAF [44]. Patients with advanced HCC without prior systemic therapy, balanced with respect to their baseline characteristics, were enrolled in SHARP. They were grouped into two cohorts and underwent either oral sorafenib (400 mg twice a day) or placebo during a radiological and clinical follow-up. The study showed a better overall survival (10.7 vs. 7.9 months; *p* < 0.001) and longer time to radiological progression (5.5 vs. 2.8 months; *p* < 0.001) in the sorafenib group. However, sorafenib could not delay the time to symptomatic progression [43].

Bruix et al. conducted a sub-analysis of a phase III trial in 2012. They suggested that sorafenib determined a longer OS and disease control rate (DCR) compared to placebo in the case of advanced HCC. This was regardless of the disease nature, previous tumor burden, performance status, stage of the tumor, or previous therapy. They included 602 patients with advanced HCC and Child–Pugh A. The patients were sorted to take either oral sorafenib at 400 mg or placebo. Sorafenib consistently determined a longer median time to progression (TTP) (HR, 0.40–0.64), except for in HBV-positive patients (HR, 1.03). Therefore, sorafenib is considered as the standard treatment for BCLC stage C tumors [45].

Yada et al. evaluated which indicators of the efficacy of sorafenib could be considered as reliable for patients with advanced HCC. They included 46 patients and considered the following variables of tumor and patient: etiology, size of tumor, location (intra or extrahepatic), and sorafenib adverse events, such as diarrhea or hand-foot syndrome (HFS). They focused on OS and TTP. The results stated that the etiology of HCC did not have a correlation with the median OS and TTP in a multivariate analysis. Extrahepatic major nodules (HR = 0.36, *p* < 0.01) and HFS (HR = 0.44, *p* < 0.05), instead, prolonged the TTP [46]. Cho et al. corroborated the thesis that patients undergoing sorafenib, with HFS and diarrhea, showed a better OS than patients not affected by complications [47].

Biomarkers for the prediction of sorafenib’s therapeutic effect were shown in a study from 2021. Yu et al. analyzed a total of 23 patients: 7 (30.4%) with complete/partial response (CR/PR); 7 (30.4%) with stable disease (SD); and 9 (39.1%) with progressive disease (PD). The chromosome 7q, presenting the multidrug resistance gene ATP Binding Cassette Subfamily B Member 1 (ACBC1), was correlated with a poor OS (*p* = 0.004) and TTP (*p* < 0.001). They concluded that sorafenib resistance in advanced HCC is due to the percentage of genome changes and the amplification of chromosome 7q [48]. Dickkopf-1 (DKK1) is usually overrepresented in HCC in comparison with controls and patients with other hepatopathies (*p* < 0.05). Seo et al. published a study in 2023 in which they evaluated the correlation of the inhibition of DKK1 with the antitumor effect of sorafenib. They found that DKK1 inhibition determines the antitumor effect of sorafenib through the inhibition of the PI3K/Akt and Wnt/β-catenin mechanisms through the regulation of GSK3β activity [49].

### 4.2. Transarterial Chemoembolization (TACE)

Several studies have demonstrated that TACE seems to have a potential role in treating cases with advanced-stage HCC [50,51,52,53,54]. Chung et al., in 2011, demonstrated how the application of TACE (hazard ratio, 0.263; *p* < 0.001) and a Child–Pugh class A status (*p* = 0.004) were independent predictive factors for an advantageous result. TACE presented promising survival benefits when matched with supportive care in patients with Child–Pugh class A and class B disease (overall survival: 7.4 months vs. 2.6 months, *p* < 0.001; 2.8 months vs. 1.9 months, *p* = 0.002, respectively) [55].

Further studies were conducted by Yoo et al. on a cohort of 251 patients with advanced HCC. They demonstrated that, regardless of the use of sorafenib, TACE offered promising survival benefits over conservative treatments in cases with advanced HCC and extrahepatic localization [56]. Other studies showed how hypoxia induced by TACE results in the release of angiogenic growth factors in surviving tumor cells, which determines tumor recurrence, extra-hepatic localization, and worse outcomes [57].

### 4.3. TACE + Sorafenib

A comparative study on patients with advanced HCC treated with sorafenib alone or sorafenib and TACE was published in 2013. It was demonstrated that the average TTP and OS were better in the case of combined therapy than in the monotherapy group (TTP: 2.5 months vs. 2.1 months, *p* = 0.008; OS: 8.9 months vs. 5.9 months, *p* = 0.009).

The combination of TACE and sorafenib has, thus, a reported benefit in the delay of tumor progression in cases with advanced HCC, but its advantage in OS is not clear [58].

### 4.4. Radiotherapy (RT) and Combined Therapies

A single-center retrospective study in 2015 enrolled 557 cases of HCC with portal vein tumor thrombosis (PVTT) [59]. The efficacy of TACE with and without RT was evaluated and compared to sorafenib. The TACE + RT group presented a longer TTP and greater OS compared with TACE or sorafenib (*p* < 0.001). A multivariate Cox analysis stated that the TACE + RT protocol was an independent predictor of a favorable TTP and OS. In a matched cohort with 102 pairs, the TTP and OS were better in the TACE + RT group than those in the TACE group (TTP, 8.7 months vs. 3.6 months, respectively, *p* < 0.001; OS, 11.4 months vs. 7.4 months, respectively, *p* = 0.023). The analysis with sorafenib with 30 pairs showed similar results (TTP, 5.1 months vs. 1.6 months, respectively, *p* < 0.001; OS, 8.2 months vs. 3.2 months, respectively, *p* < 0.001). The analysis demonstrated that TACE plus RT is a reliable and safe alternative to conventional therapy with sorafenib for management in the case of a diagnosis of advanced HCC with PVTT [59].

Sapisochin et al. published a study in 2017 about stereotactic body radiotherapy (SBRT) in bridging advanced HCC [60]. It was compared to TACE and RFA. SBRT works by hurting the DNA inside the tumor cells, like other forms of radiation therapy. The damage from the radiation stops the cells from proliferating. This causes tumors to shrink. According to their analysis, more than 370 cases were treated with RFA (n = 244), TACE (n = 99), or SBRT (n = 36). The registered rates of drop-out were similar among the groups (16.8%, 20.2%, 16.7%, respectively; *p* = 0.7). Postoperative complications were analogous, too. The cohort of the RFA group presented a higher rate of tumoral necrosis in the explanted liver. Also, the 1-year patient survival from enrolment on the transplant waiting list was 86% in the RFA group, 86% in the TACE group, and 83% in the SBRT group (*p* = 0.4). The 3-year patient survival was 72% in the RFA group, 61% in the TACE group, and 61% in the SBRT group (*p* = 0.4). The 5-year survival was 61% in the RFA group, 56% in the TACE group, and 61% in the STBR group (*p* = 0.4). The survival 1, 3, and 5 years after the transplant was also very similar in the groups. Then, it was concluded that SBRT is a good option as an alternative to conventional therapies [60]. Bauer et al. published a multi-center study in 2021 [61]. They analyzed the microscopic response to TACE and SBRT combined therapy in comparison to TACE or SBRT alone for the treatment of HCC. TACE was used in 14 cases, SBRT was applied to 4 patients, and combined therapy was used in 9 cases. The three groups did not differ in gender, age, the cause of underlying hepatopathy, and the number and size of tumor nodules. Almost all the cases in the TACE + SBRT combination group (89%) did not present residual vital tumor tissue in the liver explant. Nevertheless, TACE or SBRT as a single approach presented lower rates of histopathologic CR (0% and 25%, respectively; *p* < 0.001). These data revealed that combined TACE and SBRT improve the outcomes in bridging to OLT [61].

Wang et al. considered the radiological response and adverse events of SBRT (such as radiation-induced liver disease, RILD) in the same year. SBRT provided a response rate of 62.5%, with a 100% infield control. The worst toxicity comprised hyperbilirubinemia and thrombocytopenia. One patient developed non-classic RILD. The patients underwent a bridge to OLT with a median time of 8.4 months after SBRT, and a complete pathologic response in 23.1%. The median OS and RFS were 37.8 months and 18.3 months from the time of OLT, respectively. They concluded that SBRT determines good tumor control and tolerable adverse effects in the case of patients listed for OLT [62]. Recently, Yang et al. analyzed the effects and adverse reactions in the case of combined TACE and intensity-modulated radiotherapy (IMRT) with sorafenib in the treatment of advanced HCC with macrovascular invasion. They analyzed 82 patients: 47 were treated with the combined therapy of TACE, IMRT, and sorafenib; 35 were treated with TACE plus IMRT. The median PFS was 17.2 months (95% CI 14.1–19.9) in the TACE + IMRT + sorafenib cohort. This was significantly better than the survival observed in the other group (9.4 months; 95% CI 6.8–11.2; *p* < 0.001). In addition, those undergoing treatment with the TACE + IMRT + sorafenib protocol demonstrated a better OS than those treated with TACE plus IMRT alone (24.1 vs. 17.3 months; *p* < 0.001). Moreover, the rates of grade 1 to 2 HFS, diarrhea, hair loss, and other skin reactions were higher in the TACE + IMRT + sorafenib cohort (*p* < 0.05). They concluded that the protocol with TACE + IMRT + sorafenib is safe and offers considerable clinical results in the treatment of HCC with macrovascular invasion, improving the tumor response rate and enlarging both PFS and OS [63].

### 4.5. Immunotherapy

Immunotherapy is a fundamental paradigm in cancer treatment. Its mechanism of action is to induce an immune reaction against antigen-bearing tumor cells. The subsequent cell death releases secondary (non-targeted) tumor antigens. The secondary antigens generate a further immune response (antigen spread) [64]. Wehrenberg-Klee et al. described, in 2018, a case of advanced HCC brought to hepatic resection thanks to a successful combination of Nivolumab and TARE with Y90. The histology confirmed the complete response with negative margins. Therefore, it was hypothesized that this combination therapy may enhance the immune response to HCC [65]. Another case report was published in 2019. It described a case of HCC with the involvement of both the right and left portal vein and lung and bone metastasis. The neoplasia showed a good response after consecutive treatment with TARE, sorafenib, and nivolumab, suggesting a possible synergistic effect of these drugs on the response to HCC [66]. Nunez et al. conducted a study in 2021 demonstrating that a low lymphocyte concentration or high presence of PD-1 checkpoint inhibitors is correlated with an incomplete response to liver-directed therapy (LDT) and an increased risk of tumor progression in the bridge-to-transplant wait. PD-1 immunotherapy could be beneficial for patients with impaired T cell homeostasis in order to increase the response to LDT and bridge-to-transplant outcomes [67]. A study published in 2023 demonstrated that the treatment of advanced HCC can exploit several FDA-approved drugs such as immune checkpoint inhibitors (ICIs) with or without anti-angiogenics and tyrosine kinase inhibitors. According to this study, almost 66% of patients do not benefit from ICIs. The actual ongoing protocol with CAR-T cell therapy, directed toward other checkpoint molecules such as TIM-3, LAG-3, and TIGIT, will enforce ICI-based therapeutic possibilities for advanced HCC. Thanks to further research, small-molecule inhibitors influencing the PD-1/PD-L1 pathway can create a good alternative associated with a better oral bioavailability, antitumor efficacy, and less adverse effects [19].

## 5. Discussion and Conclusions

Sorafenib is still the recognized treatment for advanced HCC, with an effective increase in OS and a low rate of drug toxicity [68]. In addition, survival with sorafenib is reported to be less than 3 months, with a need for alternative therapeutic strategies [19].

An increasing rate of tumor recurrence and metastases is also due to TACE-induced hypoxia in surviving tumor cells, thanks to the release of angiogenic growth factors. This determines a lower survival rate. Sorafenib exerts an antiangiogenic effect and a consequent inhibition of tumor cell proliferation by blocking the Raf-MEK-ERK pathway. Therefore, it is thought that a protocol with TACE and sorafenib might be beneficial in HCC [69,70,71]. This combination has a reliable effect on inhibiting tumor progression in the case of advanced HCC, without a clear survival benefit.

A few studies have been conducted on evaluating the effectiveness of RT. These showed that patients undergoing TACE + RT have better outcomes than those undergoing TACE or sorafenib alone in terms of OS and TTP. The effectiveness of RT, TACE, and RFA was then evaluated, and it was observed that the dropout rate was comparable in all the three cases. A recent study also evaluated the combined therapy of TACE + RT + sorafenib compared to the association of TACE + RT in terms of progression-free survival, demonstrating better results in the first group.

Immunotherapy also generates new perspectives for the treatment of advanced HCC. In fact, it has been observed that patients with a low lymphocyte count can obtain a better response to LDT from PD-1 immunotherapy. The effectiveness of immuno-mediated therapy holds the promise of positively influencing bridge-to-transplant outcomes. Moreover, oncological downstaging could be so consistent as to increase the number of patients meeting the transplantability criteria. This could lead to a clear improvement in post-OLT and bridge-to-transplant outcomes [19]. However, it will be necessary to develop dedicated post-transplant immunosuppressive management protocols in the event of exposure to immuno-therapeutic drugs.

Even if this review shows significant results, we are aware it has some limits. First of all, some studies had consistent results, but the studied population consisted of a small cohort. Moreover, the cohorts in the different articles were heterogenous. In fact, the patients who were analyzed had different grades of disease and comorbidities. In addition, the proposed treatments were varied, too. But, if this is a recognized limit for a study, it is also true that it can make the results even more useful for clinicians. As a matter of fact, it reflects the “real-life” scenario.

Considering all this variability, further prospective and multi-centric studies are needed for validating the results. The introduction of new strategies, such as immunotherapy, requires more and more multidisciplinary approaches also including immunologists. This would allow for a patient-“tailored” strategy. In the next few years, artificial intelligence could also help in selecting the best available option for each patient according to their characteristics and the peculiarities of the disease, as already shown in some studies [72,73]. Artificial intelligence would also give a prediction of survival and recurrence risk for each treatment. Furthermore, even if there is growing interest in it, artificial intelligence is still not available for routine clinical practice. More experiences will be needed before its potential and substantial application.

## Figures and Tables

**Figure 1 medicina-60-01010-f001:**
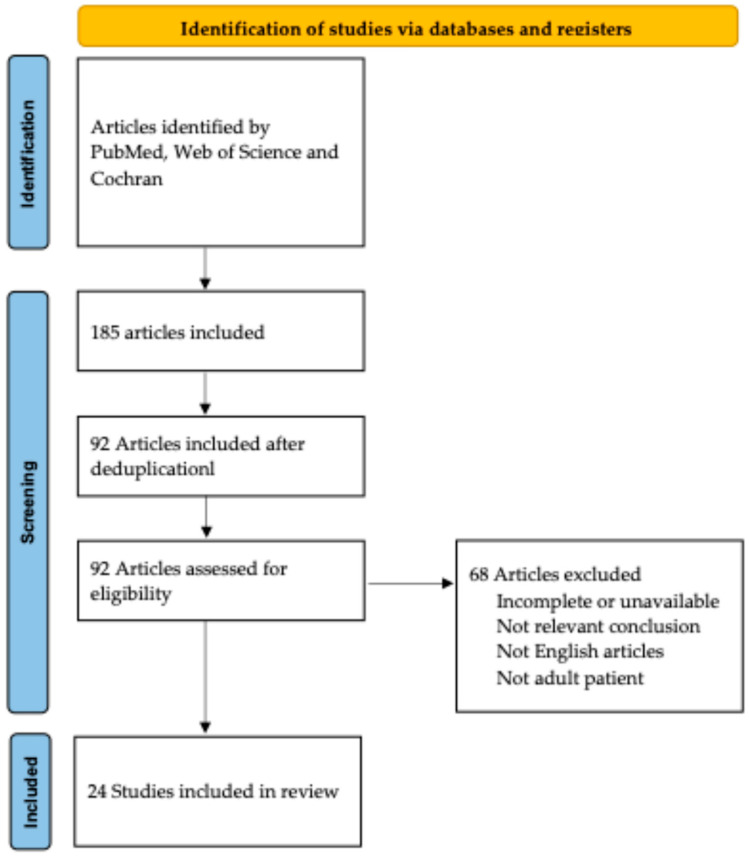
PRISMA flow diagram.

**Figure 2 medicina-60-01010-f002:**
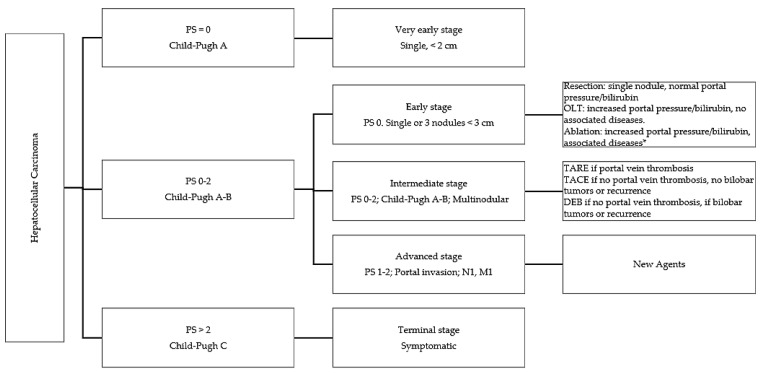
Proposed treatment for different stages of HCC according to Child–Pugh grade. * Cryoablation if 3 nodules <3 cm with perivascular invasion and poor anesthesia candidate; MVA if 3 nodules <3 cm with perivascular invasion and good anesthesia candidate if 3–6 cm; and RFA if <3 cm and no perivascular invasion. OLT: orthotopic liver transplant; TARE: rransarterial radioembolization; TACE: rransarterial chemoembolization; and DEB: drug-eluting beads.

**Figure 3 medicina-60-01010-f003:**
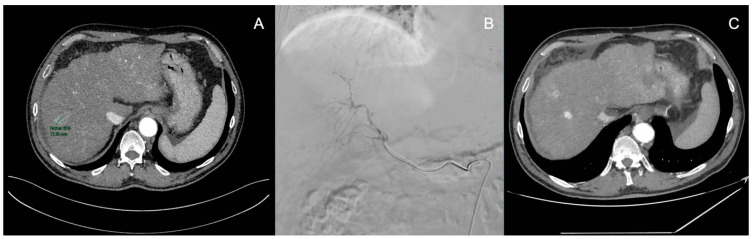
HCC nodule treated through TACE (**A**): CT scan of an HCC nodule; (**B**): TACE (B); and (**C**): re-evaluation one month after TACE (personal observation).

## Data Availability

All the data used are present in the text. No additional data are available.

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
