# Peer review of "Bridge Therapy before Liver Transplant for Advanced Hepatocellular Carcinoma"

_medicina, 2024, doi:10.3390/medicina60061010_

Round 1

Reviewer 1 Report

Comments and Suggestions for Authors

The aim of the review article sent to me for peer review is to evaluate whether bridge therapies in advanced hepatocellular carcinoma can improve overall survival and reduce the likelihood of de-listing. The study is meticulously crafted, well-organized, and flows nicely, making it an engaging read. It is evident that considerable effort has been put into synthesizing and presenting the findings in a coherent manner.

This review is expected to serve as a valuable resource for researchers working in the field of hepatocellular carcinoma, providing them with a comprehensive summary of the existing literature. The thorough analysis and discussion presented in the paper will likely help in guiding future research directions and clinical practices.

However, there is a critical discrepancy that needs to be addressed. The abstract mentions that 185 articles were selected for the review, yet the references section lists only 65 studies. This inconsistency might cause confusion for readers and needs to be resolved to maintain the integrity of the review.

It is suggested that the authors clarify this discrepancy, either by updating the references to include all 185 articles mentioned or by amending the abstract to reflect the accurate number of studies reviewed. Once this issue is resolved, the article will be a significant contribution to the literature on bridge therapies in advanced hepatocellular carcinoma.

Author Response

My coauthors and I thank the reviewers for the appreciation they have shown towards our manuscript. The manuscript has been modified taking into account the reviewers' suggestions. Text changes were written in red.

In consideration of the indications of reviewer 1, we have explained the selection of scientific publications better in the Materials Paragraph. Of the original 185 articles, after careful selection only 24 articles were used for this review.

We have included a PRISMA diagram to make this aspect better evident. Furthermore, the list of references has been modified.

Reviewer 2 Report

Comments and Suggestions for Authors

Overall, this review makes a significant contribution to the field of HCC treatment and bridging therapies. Addressing the comments below will enhance the manuscript's clarity, comprehensiveness, and impact.

Major Comments:

Significance and Contribution:

The review addresses an important aspect of HCC management, especially given the limited availability of liver transplants and the high risk of disease progression while on the waiting list. The focus on bridging therapies is highly relevant and contributes valuable insights into current clinical practices and future directions.

Literature Review and Background:

The article provides a comprehensive review of the literature, referencing key studies and criteria like the Milan and UCSF criteria. However, the review could be strengthened by including more recent studies and statistical analyses to provide updated evidence and trends in bridging therapies.

Methodology:

The methodology for article selection is clear and appropriate, focusing on adult patients and English-language studies. However, more detail on the specific inclusion and exclusion criteria for studies would enhance transparency and reproducibility.

Analysis and Interpretation:

The analysis is thorough, discussing the benefits and limitations of various bridging therapies. However, the review would benefit from a more critical analysis of the data, including potential biases in the selected studies and variability in patient populations.

Figures and Tables:

Figure 1 and 2 provide useful visual summaries of treatment strategies and outcomes. It would be beneficial to include more figures or tables summarizing key study results, such as survival rates and progression-free survival for different therapies.

Conclusion:

The conclusion effectively summarizes the findings and highlights the potential of combination therapies as a new standard. However, it would be more impactful with specific recommendations for clinical practice and future research directions.

Minor Comments:

Abstract:

The abstract is concise but could provide more specific data on the outcomes of different therapies to give a clearer snapshot of the review findings.

Introduction:

The introduction is well-written, but the flow could be improved by reorganizing some sections to better set the stage for the review. For example, discussing the significance of bridging therapies earlier could capture the reader's attention more effectively.

References:

The references are appropriate, but it would be helpful to see more recent studies included to ensure the review reflects the latest advancements in the field.

Recommendations:

Reorganize the introduction to better highlight the importance of bridging therapies early on.

Include more recent studies and statistical analyses in the literature review.

Provide detailed inclusion and exclusion criteria in the methodology section.

Add more figures or tables summarizing key study outcomes.

Enhance the conclusion with specific clinical and research recommendations.

Correct grammatical errors and improve the overall flow and clarity of the manuscript.

Comments on the Quality of English Language

The manuscript is generally well-written, but there are minor grammatical errors and awkward phrasings that should be corrected for clarity. For example, "that is why they was used" should be corrected to "that is why they were used" (line 48).

Author Response

My coauthors and I thank the reviewers for the appreciation they have shown towards our manuscript. The manuscript has been modified taking into account the reviewers' suggestions. Text changes were written in red.

In consideration of the indications of reviewer 2, we have explained the selection of scientific publications better in the Materials Paragraph. Of the original 185 articles, after careful selection only 24 articles were used for this review.

We have included a PRISMA diagram to make this aspect better evident. The exclusion criteria for scientific publications are reported. Furthermore, the list of references has been modified.

We have changed the discussion. The edited text is written in red.

We have not included tables that summarize the results of each individual procedure, because the procedures themselves are often used in combination in the individual patient. However, the most important results are reported in each individual paragraph.

The discussion and conclusions have been modified.